# Evolution of Three-Finger Toxin Genes in Neotropical Colubrine Snakes (Colubridae)

**DOI:** 10.3390/toxins15090523

**Published:** 2023-08-25

**Authors:** Kristy Srodawa, Peter A. Cerda, Alison R. Davis Rabosky, Jenna M. Crowe-Riddell

**Affiliations:** 1Ecology and Evolutionary Biology, University of Michigan, Ann Arbor, MI 48109, USA; ksrodawa@umich.edu (K.S.); ardr@umich.edu (A.R.D.R.); j.crowe-riddell@latrobe.edu.au (J.M.C.-R.); 2Molecular, Cellular, and Developmental Biology, University of Michigan, Ann Arbor, MI 48109, USA; 3Museum of Zoology, University of Michigan, Ann Arbor, MI 48108, USA; 4School of Agriculture, Biomedicine and Environment, La Trobe University, Melbourne, VIC 3086, Australia

**Keywords:** snake venom, neurotoxin, molecular evolution, gene families, opisthoglyphous

## Abstract

Snake venom research has historically focused on front-fanged species (Viperidae and Elapidae), limiting our knowledge of venom evolution in rear-fanged snakes across their ecologically diverse phylogeny. Three-finger toxins (3FTxs) are a known neurotoxic component in the venoms of some rear-fanged snakes (Colubridae: Colubrinae), but it is unclear how prevalent 3FTxs are both in expression within venom glands and more broadly among colubrine species. Here, we used a transcriptomic approach to characterize the venom expression profiles of four species of colubrine snakes from the Neotropics that were dominated by 3FTx expression (in the genera *Chironius*, *Oxybelis*, *Rhinobothryum*, and *Spilotes*). By reconstructing the gene trees of 3FTxs, we found evidence of putative novel heterodimers in the sequences of *Chironius multiventris* and *Oxybelis aeneus*, revealing an instance of parallel evolution of this structural change in 3FTxs among rear-fanged colubrine snakes. We also found positive selection at sites within structural loops or “fingers” of 3FTxs, indicating these areas may be key binding sites that interact with prey target molecules. Overall, our results highlight the importance of exploring the venoms of understudied species in reconstructing the full evolutionary history of toxins across the tree of life.

## 1. Introduction

Colubridae is a large family of snakes that includes over half of all extant snake species (>2000 out of 4000 species; [1]), of which approximately ~700 species are considered venomous to their prey [2]. Despite having comparable species richness to medically important front-fanged snake families, Viperidae and Elapidae (~700 species; [1,3,4]), the venoms of rear-fanged colubrid snakes have not been well studied until recently [5]. This gap has been attributed to the low incidence of lethal envenomation to humans, as well as the difficulties in extracting venom from the low-pressure venom system of rear-fanged snakes [6]. Given these challenges, whole transcriptome approaches can greatly assist in venom studies of colubrids by determining venom composition and variation among and between species [7,8], identifying novel putative toxins [9] and generating new comparative data for evolutionary studies [10,11,12]. Recent analysis of rear-fanged snakes has revealed novel venom components [9], simplified venom compositions [7,8,12], and prey-specific toxins [13,14,15]. However, only about a quarter of published snake venom transcriptomes are of venom glands from rear-fanged snakes [16], which limits our capacity to fully evaluate the evolutionary history and potential for novel structural changes in venom genes across snake diversity.

Three-finger toxins (3FTxs) dominate the venom profiles of several venomous colubrid snakes of the subfamily Colubrinae [6,8,12,14,15], although the extent of this pattern is currently unknown [17]. 3FTxs are small proteins that are generally non-enzymatic and bind various ligands to induce a variety of biological effects driven by structural differences, including neurotoxicity, cardiotoxicity, anticoagulation, and cytotoxicity [1,18]. This toxin superfamily was first discovered in front-fanged elapid snakes [19], where their “conventional” structure consists of eight conserved cysteines that create three β-stranded loops, or “fingers”, stabilized by four disulfide bonds [20,21]. Much of our knowledge of the evolution of 3FTxs comes from elapid snakes, which show high rates of gene duplication [22,23,24], strong positive selection [23,24], and high functional diversity [20,21]. Colubrine 3FTxs are considered “non-conventional” in structure because they feature an additional fifth disulfide bond located in loop I or II [20]. In some colubrine species, non-conventional 3FTxs target and bind to nicotinic acetylcholine receptors, causing flaccid paralysis in envenomated prey [21]. Like their elapid counterparts, colubrine 3FTxs have undergone extensive gene duplication, but the evolutionary history of this gene family across colubrine species has not been thoroughly investigated [11,13]. Further, while patterns of sequence variation have been observed in 3FTx sequences of several colubrine species, sites that are under selection that may be associated with the functional diversification of these toxins have not been extensively evaluated [11,25].

Several colubrine species have 3FTxs with taxon-specific prey toxicity enabled by heterodimeric interactions among their proteins. For example, cat snakes, *Boiga irregularis* and *Boiga dendrophilia*, have bird and lizard specific non-conventional 3FTxs, irditoxin and denmotoxin, respectively [14,15]. Despite being phylogenetically distant colubrine snakes, prey-specific 3FTxs of *Boiga* species (i.e., irditoxin and denmotoxin) and *Spilotes* species (i.e., sulditoxin) form the same linked heterodimers enabled via disulfide bonds between additional cysteine residues in the first and second loops of the alpha and beta subunits [13,14,15]. Similar additional cysteines have been discovered through RNA-sequencing studies of other venomous colubrines, such as *Boiga nigriceps*, *Telescopus dhara*, and *Trimorphodon biscutatus*, which suggest that these species also have heterodimeric 3FTxs [26]. Our current understanding is that the heterodimeric interaction has evolved independently in *Boiga* species as compared to other colubrine snakes, but the extent of this parallel evolution of heterodimeric interactions between 3FTxs is not fully known due to a lack of data across broadly sampled taxa.

We used a transcriptomic approach in Neotropical rear-fanged colubrine snakes to determine (1) how widespread the 3FTx venom profile may be, (2) whether positive selection is driving 3FTx differentiation, and (3) whether there are other 3FTx capable of forming novel heterodimeric interactions. We sequenced the venom gland transcriptome of individuals of four colubrine species: *Chironius multiventris*, *Oxybelis aeneus*, *Rhinobothryum bovallii*, and *Spilotes sulphureus*. Of the four species studied in this paper, three have had previous venom proteomic or transcriptomic studies where 3FTxs were the most abundant protein family [13,27,28]. *C. multiventris* had no previous comparable studies. Based on previous studies, we predicted that 3FTx would dominate the toxin expressions of these colubrine species and allow for investigation of the variations in amino acid structure and selection patterns across key toxin sequences.

## 2. Results and Discussion

### 2.1. Colubrine Venom Transcriptomes Are Dominated by 3FTxs

Our transcriptomic approach generated venom gland transcriptomes for individuals of four Neotropical colubrine species: *C. multiventris*, *O. aeneus*, *R. bovallii*, and *S. sulphureus* (Table 1). The toxin transcriptomes included putative toxin genes from multiple venom superfamilies in addition to non-conventional 3FTxs, including cysteine-rich secretory proteins (CRiSP), Kunitz-type serine protease inhibitor (Kunitz), phospholipases B (PLB), phospholipase A_2_s (PLA_2_s), snake venom metalloproteinases (SVMPIIIs), snake venom serine proteases (SVSPs), and Waprin (Table 1 and Appendix A). No conventional 3FTxs were found among the five specimens. While many toxin transcripts were recovered, non-conventional 3FTxs dominated the expression profiles in all of the rear-fanged colubrines despite the differences in the shape and size of venom glands and fangs (see Figure 1 and Figure 2). 3FTx expression represented 93–99% of all toxin transcript expression, with the top three to five transcripts expressed being 3FTxs (Figure 2). The highest-ranking 3FTx transcript had more than one order of magnitude greater relative abundance (FPKM) than the highest-ranking non-3FTx transcript (Figure 2). The highest ranking 3FTxs had TPM values of 777,346 (*O. aeneus*), 735,857 (*S. sulphureus,* 643,746 (*C. multiventris* B), 571,608 (*C. multiventris* A), and 369,994 (*R. bovallii*), as compared to the highest ranking non-3FTxs with TPM values of 5636 (*O. aeneus,* KUNITZ family), 5288 (*S. sulphureus,* CRISP family), 29,444 (*C. multiventris* A, CRISP family), 35,338 (*C. multiventris* B, CRISP family), and 5659 (*R. bovallii*, CRISP family) (Appendix A).

The simple venom gene expression of 3FTx profiles of our snakes (Appendix A) are consistent with transcriptomic and proteomic data in previously analyzed conspecifics [13,28] and other rear-fanged colubrines [7,8,29,30]. Studies of *Oxybelis* species show a simple venom proteome comprising 3FTXs (including fulgimotoxins from *Oxybelis fulgidus*), SVMP III, L-amino acid oxidase, and CRiSP [27]. Although we recovered members of SVMP III, L-amino acid oxidase, and CRiSP from our sample of *O. aeneus*, the majority of toxin transcripts produced by this individual were 3FTxs (Figure 2). The toxin transcriptome of *R. bovallii* recovered here was dominated by 3FTx (95%) followed by CRiSP (1.5%) and PLA_2_ (1.2%) and a smaller percent of SVMPIII; similarly, the venom proteome of *R. bovallii* is dominated by 3FTx (86.5%) followed by CRiSP (8.2%) and SVMP III (2.4%) [28]. Previous transcriptome profiling of *S. sulphureus* venom glands also found that 3FTxs were highly expressed, albeit at a lower percentage compared to this study (60% vs. 99%), and as a large portion of the total venom gland proteome (92% 3FTx, 6% CRiSP, 1% SVMP III; [13]). Approximately 1% of our *S. sulphureus* transcriptome expression recovered CRiSP, PLA_2_, and SVMP. Venom transcriptomes were consistent in both individuals of *C. multiventris* showing highest expression of 3FTx (93–94%), followed by CRiSP (~5%) and lower expressions of C-type lectins, SVMP, PLA_2_, Waprin, PLB, SVSP, and Kunitz (~0.9% total). We could not compare the venom gland transcriptome of *C. multiventris* because there were no proteome or toxicity data currently available, but their 3FTx-dominated toxin expression was very similar to some other colubrines [8,12,13,25].

Although geographic variation is a common phenomenon in snake venoms [31] our results support a uniform venom profile within *R. bovallii* and *S. sulphureus* colubrine snakes. Our specimens were collected in Nicaragua and Peru, respectively, whereas the *R. bovallii* analyzed by Calvete et al. [28] were wild caught in Costa Rica, and the *S. sulphureus* individual studied by Modahl et al. [13] was a captive specimen originally sourced from Suriname. Lack of geographic variation in venom types has been found before in elapid species that use neurotoxic venom components [32], but this pattern would need to be confirmed in colubrid species examined here by examining the toxin expression of more individuals sampled throughout their South American range.

3FTxs are functionally diverse among and within snake venoms, causing numerous biological effects in prey, such as neurotoxicity or cardiotoxicity, but much of our knowledge is based on elapid snake toxins [13,24,33]. The broad effects of 3FTxs are due to mutations in the binding site, which alter target specificity [21]. Currently, our understanding of the functions of colubrine 3FTxs is that they bind and block nicotinic acetylcholine receptors (nAChRs) in skeletal muscles, causing flaccid paralysis [14,15,26]. It is unknown if other functional characteristics exist among colubrine 3FTx, but this would be an exciting future research target. Furthermore, individual 3FTxs may prove to become more effective in different taxa, resulting in taxon-specific toxicity in 3FTxs [13,14,15]. These types of toxins have already been described in several colubrine snakes, including *S. sulphureus* [13,14,15]. Given the taxon-specific nature of several *S. sulphureus* 3FTxs [13], functional and prey toxicity studies should also be performed on the venoms of Neotropical colubrine snakes to determine if variation in functional toxicity exists.

### 2.2. Parallel Evolution of Heterodimeric Sequences

The evolutionary history of 3FTxs across taxa is shown using gene trees from the putative 3FTx transcripts recovered from our colubrine transcriptomes and 3FTx sequences currently available from several other colubrines and elapids, with a single viper species as an outgroup (Figure 3). The tree topology was generally consistent between Bayesian (Figure 3) and maximum likelihood methods (Appendix A). Colubrid sequence relationships remained constant between the two trees, with the exception being that the major polytomy within colubrine sequences found in the Bayesian tree were weakly resolved in the maximum likelihood tree (Appendix A). As the topologies were similar between the two methods, we focused on the Bayesian tree. We recovered elapid sequences as sister to all the colubrine sequences, as expected from previous studies of 3FTxs [11,13] (Figure 3). Our inability to resolve a large polytomy is possibly due to the short sequence length and rapid rate of evolution, which has impacted other studies of the evolutionary history of 3FTxs [23,34]. We found that sequences from the same individual or species often clustered together (Figure 3), suggesting that lineage (species)-specific duplication events have occurred. Gene duplication plays an important role in the evolution and diversification of snake toxin gene families [23,35,36], yielding material for mutation and selection to act on, which can potentially diversify toxin function via neofunctionalization [37,38]. Evidence of gene duplication has been found in several colubrine species, but these studies have largely been transcriptomic [11,13,39] and rely on the capture of expressed toxin sequences. To determine if gene duplication is driving 3FTxs diversity, a genomic approach should be taken to uncover the full repertoire of 3FTx loci in colubrines.

The tree topology of colubrine 3FTxs shows evidence for putative homologous sequences with 3FTx that have known taxon-specific toxicity in other rear-fanged colubrines (in blue—Figure 3). Of the 31 putative sequences of interest, 30 were full coding sequences (partial coding sequence *C. multiventris* B 3FTx-1; Figure 4). Some transcripts from our *S. sulphureus* clustered with known taxon-specific 3FTxs previously identified for the same species: sulditoxin A (transcript ID: 3FTx-5) and sulmotoxin-1 (transcript ID: 3FTx-4), but not sulditoxin B (Figure 3). It is possible that the *S. sulphureus* individual studied here was simply not expressing sulditoxin B at the time of death or that we were unable to recover the transcript. None of our sequences clustered in the same clade as the taxon-specific 3FTxs of *Boiga* cat snakes (i.e., denmotoxin, irditoxin A, or irditoxin B). We did not recover any transcript from our *O. aeneus* transcriptome that clustered near fulgimotoxin from closely related *O. fulgidus*, suggesting that fulgimotoxin may be independently derived and unique to *O. fulgidus*. However, greater inter- and intraspecific sampling of *Oxybelis* species should be performed to test these possible conclusions.

We determined the presence of putative heterodimeric 3FTxs in our samples that are unique to other colubrine 3FTxs by translating complete 3FTx coding sequences from our transcriptome and aligned with amino acid sequences of 3FTxs that have known taxon-specific toxic effects (i.e., sulditoxin A and B and sulmotoxin-1, irditoxin A and B, and denmotoxin) [13,14,15] (Figure 4). We annotated β-stranded loops or “fingers” according to conserved patterns of cysteines that create the five distinctive loops in non-conventional 3FTxs, showing that the amino acid structures of colubrine 3FTxs appear to be highly conserved across species (Figure 4). However, five recovered transcripts lacked a cysteine for Loop I (Figure 4). This suggested that these sequences may be nonfunctional, as they likely lack the ability to form the non-conventional 3FTx structure.

The translated amino acid alignment indicated extra cysteine residues in *C. multiventris* and *O. aeneus*, parallel to 3FTx in other colubrines with known taxon-specific toxic effects. Sulditoxin A/B and irditoxin A/B possess extra cysteine residues at positions 59 and 86 (Boxes—Figure 4), respectively, which are known to be a part of the heterodimeric interaction between A and B subunits. Some sequences from our *C. multiventris* transcriptomes have additional cysteines (Boxes in Figure 4) in the same positions as both irditoxin and sulditoxin A and B subunits, suggesting that the secondary protein structure enables heterodimer formation. The *C. multiventris* sequences cluster closely with their sulditoxin A and B counterparts in the gene tree (Figure 3), implying that this heterodimeric interaction is conserved among members of this clade of colubrine snakes. Additionally, the *O. aeneus* sequence, which also contains an extra cystine residue, groups closely with sulditoxin A and putative *C. multiventris* heterodimeric sequences. Indeed, predicted 3D protein structures and functions for the colubrine 3FTxs consistently matched the two subunits of heterodimer-forming irditoxin [14]. More nuanced differences in protein structures among sequences, however, are unlikely to be predicted due to the limited number of published non-conventional 3FTx protein structures [11,25]. The potential for heterodimeric formations among *C. multiventris* sequences suggest the 3FTx protein products may have taxon-specific toxicity, similar to heterodimeric sulditoxins (which are lizard-specific [13]) and irditoxin (which are bird- and lizard-specific [14]). Indeed, *C. multiventris* is a semi-arboreal species with diet records of lizards and frogs [40]. Proteomic studies of venoms of *C. multiventris* are needed to determine if each 3FTx subunit forms a covalent heterodimer and, if so, whether this functional state is involved in taxon-specific toxicity for lizard/frog prey.

Our results further support the notion that 3FTx heterodimeric interactions evolved independently from the similar structure and interaction observed in *Boiga* cat snakes [11,25]. An Indomalysian–Australasian lineage within the colubrinae radiation, *Boiga* cat snakes are phylogenetically distinct from the Neotropical lineages of *Spilotes* and *Chironius*, which are more closely related to each other than *Boiga* [3]. Other species with putative heterodimeric 3FTxs cluster within the same clade as either *Spilotes/Chironius* (i.e., *Trimorphodon, Oxybelis*) or *Boiga* (i.e., *Telescopus*). This phylogenetic spread suggests that 3FTx heterodimeric interactions evolved independently at least twice across the radiation of rear-fanged colubrids. However, the dearth of venom sequence data across the phylogenetic breadth of rear-fanged colubrids precludes testing this hypothesis. Further studies should aim to characterize 3FTx diversity among colubrids, especially in species with divergent prey preferences, to test the parallel evolution of heterodimeric interactions for targeting taxon-specific prey.

### 2.3. Colubrine 3FTxs Are under Positive Selection

We used a range of selection tests to determine the type and strength of selection on codon sites in colubrine 3FTx transcripts. All CodeML model comparisons were found to be significant, indicating these sequences were under positive selection (ω > 3.0; Table 2). We found evidence of positive selection operating at numerous sites, particularly within loop II, (Table 3; Figure 4). In previously studied transcriptomes of *S. sulphureus*, Modahl et al. [13] observed that loop II was highly variable among 3FTx transcripts, but they did not test if selection was occurring in this region. Signatures of positive selection in loop II suggest that this area may be a main binding site for target nicotinic acetylcholine receptors in prey muscles [21]. We also found evidence of positive selection within other loops, especially in loop V, whereby all sites within the loop were under selection. Unsurprisingly, we found all cysteine codon sites to be under negative (purifying) selection as they were highly conserved across taxa sequences and vital for stabilizing the distinctive β-stranded loops of 3FTx. Selection tests on individual sites are summarized in Appendix A.

We identified several sites under positive selection located in the long N-terminus region. Previously, Xie et al. [11] found two types of N-terminus regions within 3FTx sequences recovered from rear-fanged snakes: a long region that appeared to be widespread in colubrine snakes and a short region restricted to *Boiga* species. All but three of the transcripts we recovered contained sequences for peptides with the long N-terminus (Figure 4). We did not recover any sequences that expressed a short N-terminus region, suggesting that this feature might be unique to *Boiga* species. As this region is under positive selection and is retained across several species, it may serve as a functional venom component [11]. Proteomic studies and functional tests should be performed, however, before confirming that this region is involved with the envenomation process.

## 3. Conclusions

Our analysis of several colubrid snake venom gland transcriptomes revealed high levels of 3FTx expression across several colubrine species. We identified several putative heterodimeric 3FTx sequences based on additional cysteine residues and posit these as evidence of parallel evolution of this putative subunit interaction. Our selection tests points to regions within 3FTx sequences with evidence of positive selection for sites that interact with prey targets and negative selection for sites of structural importance. Our results support evidence of parallel evolution among 3FTx in colubrine venoms, but protein purification and crystallography are needed to determine if additional cysteine residues enable the formation of a heterodimer. Furthermore, functional studies of the heterodimeric proteins should be performed on potential prey types to determine taxon-specific toxicity to prey with different physiologies.

## 4. Materials and Methods

### 4.1. Specimen Selection

Venom glands were excised for RNA-sequencing from *Chironius multiventris* (2 individuals), *Oxybelis aeneus* (1 individual), *Rhinobothryum bovallii* (1 individual), and *Spilotes sulphureus* (1 individual). Except for *S. sulphureus* [13], nothing is known of the venom gland transcriptomes of these species. We included *S. sulphureus* to ensure that our sequencing and transcriptome profiling were effective (i.e., toxins recovered from *S. sulphureus* would group together with previous studies) and whether geographic variation might influence venom expression in this species. The specimens were collected during field trips to Peru and Nicaragua (Table 4). We euthanized captured snakes following approved protocols from the University of Michigan Institutional Animal Care and Use Committee (Protocols #PRO00006234 and #PRO00008306), the Servicio Nacional Forestal y de Fauna Silvestre in Peru (SERFOR permit numbers 029-2016-SERFOR-DGGSPFFS, 405-2016-SERFOR- DGGSPFFS and 116-2017-SERFOR-DGGSPFFS), and the Ministerio del Ambiente y los Recursos Naturales in Nicaragua (MARENA permit numbers DGPNB-IC-19-2018 and DGPNB-IC-20-2018). Briefly, we euthanized snakes using a two-step protocol, with an intracoleomic 0.5 mL dose of Xylazine (at 20 mg/mL) followed by a 0.5–1 mL intracardiac injection of a saturated Chlorobutanol solution after the animal lost its righting reflex. We measured the masses, lengths, sexes, and approximated ages of the specimens. All venom glands were extracted, preserved in RNALater (Invitrogen, Carlsbad, CA, USA), and exported to the University of Michigan, whereas specimens were accessioned at both the University of Michigan Museum of Zoology (UMMZ) and Museo de Historia Natural de la Universidad Nacional Mayor de San Marcos (MUSM; Table 4).

### 4.2. MicroCT Scanning

To visualize the morphology of the venom system, we scanned a representative specimen from each species using a Nikon Metrology XTH 225ST microCT scanner (Xtect, Tring, UK) at the UMMZ. To enhance soft tissue contrast, we stained specimens in 1.25% Lugol’s iodine solution following protocols for diffusible iodine contrast-enhanced computed tomography (diceCT) in snakes [41]. We segmented the fang-bearing maxillary bone and venom gland in Volume Graphics Studio Max version 3.2 (Volume Graphics, Heidelberg, Germany) using the “threshold” and “draw” tools. We noted the position on the maxilla and extent of grooving on the fangs. Figures were prepared using Adobe Illustrator v24.0.2 (Adobe Inc., San Jose, CA, USA).

### 4.3. RNA Extraction and Transcriptome Assembly

We extracted venom gland RNA using recommended protocols of the PureLink RNA mini kit (Life Technologies, Carlsbad, CA, USA) and following Cerda et al. [6]. We submitted extracted RNA to the University of Michigan Advanced Genomics Core, where quality was assessed, and cDNA libraries were constructed and sequenced on an Illumina NovaSeq 6000. We evaluated the quality of raw sequences using FastQC v0.11.6 [42]. We used Trimmomatic v.0.36 to remove adapter sequencers and low-quality reads [43,44]. We created a de novo transcript sequence reconstruction from RNA-seq using the Trinity v2.6.6 platform for reference generation and analysis [44,45].

### 4.4. Gene Identification and Abundance Estimate

We used TransDecoder v1.0.3 to find open reading frames and then annotated our sequences using BLASTp against known protein coding genes in the Uniprot database [44,46]. We used CD-HIT [47] to cluster similar sequences and RSEM v1.2.28 [48] and Bowtie v2.3.4.1 [49] to estimate the abundance of genes expressed and calculated Fragments Per Kilobase Million (FPKM) and Transcripts Per Million (TPM). We used FPKM to compare transcript abundance within individuals, as the values as the sum value were unique to each individual, and used TPM to compare transcript abundance among individuals, as the sum total was standardized at one million across samples [48,49]. We created a custom nucleotide sequence database by downloading venom gland transcriptomes and snake venom protein sequences from NCBI GenBank [8,50,51,52,53,54]. We used this custom database to identify which sequences were toxins in our transcriptomes via BLASTn [46]. We used R [55] to write a custom script to annotate our nucleotide sequences, identify toxin gene family, and pair sequence identity with transcript abundance estimates.

### 4.5. Gene Tree Construction

To evaluate the relationships of non-conventional 3FTx sequences, we aligned our putative 3FTx sequences with 3FTx sequences from other rear-fanged species (Colubridae: Colubrinae) as well as representatives from front-fanged families (Elapidae and Viperidae) recovered from GenBank (Accession numbers listed in Appendix A). We aligned sequences with the Multiple Sequence Comparison by Log-Expectation (MUSCLE; [56]) via the plugin in Geneious Prime v.2021.0.3 [57]. We checked all alignments for ambiguities by eye and removed erroneous sequences. The nucleotide alignment contained 86 total sequences, of which 22 were our newly generated sequences. To identify conserved cysteines responsible for disulfide bonds that create the distinctive loops or “fingers” of 3FTx sequences, we translated nucleotide alignments from this subset using the “translate” function and MUSCLE plugin in Geneious. We selected the appropriate partitioning schemes and best-fit models using PartitionFinder v2.1.1 [58] with branch lengths linked and utilization of the greedy search algorithm. We built gene trees using Bayesian analyses with the MrBayes v.3.2.6 [59] plugin in Geneious, which used Markov chain Monte Carlo methods to estimate the posterior distribution of model parameters. We ran for one million generations, sampling every 1000 generations with the first 10% of sampled trees discarded as burn-in with lset rates set to “invgamma”. We also generated a maximum likelihood gene tree using the RAxML plugin in Geneious with 1000 non-parametric bootstrap replicates. We set a single sequence from *Protobothrops mucrosquamatus* (GenBank accession number XM_015823122.1) as a viper outgroup sequence for the colubrine and elapid sequences.

### 4.6. Protein Prediction

To predict secondary and tertiary structure of putative toxins transcripts, we submitted translated amino acid sequences to the I-TASSER Protein Structure and Function Predications online server (https://zhanggroup.org/I-TASSER/ (accessed on 16 August 2023; [60]). This server used a deep-learning approach by first identifying structural templates from Protein Data Bank (PDB) and then deriving functional insights by re-threading the 3D models through the protein function database BioLiP [60,61,62].

### 4.7. Selection Tests

To test for selection acting on 3FTx transcripts recovered from species examined here, we used the 3FTx MUSCLE alignment and Bayesian gene tree of our sampled 3FTx sequences as inputs for easyCodeML [63], which tested models of selection acting on sites [64,65]. We made comparisons of three pairs of site models to determine the type and strength of selection acting on specific codon sites. The first comparison was M0 (one ratio), which can determine whether selection is acting on a sequence, and M3 (discrete), which is constrained to determine whether neutral evolution or negative selection are acting on sites. The second comparison was of M1 (neutral) and M2 (selection), in which M2 allows for positive selection to act on the sites. Finally, we compared models M7 (beta) and M8 (beta and omega), which are similar to M1 and M2, respectively. However, the two pairs of models differs in that M7 and M8 allows for there to be a distribution of beta values. We also used the webserver datamonkey.org (accessed 15 December 2022) to conduct two analyses implemented in HyPhy 2.5 [66]: Mixed Effects Model of Evolution (MEME; [67]) and Fast Unconstrained Bayesian Approximation (FUBAR; [68]). MEME tests for evidence of episodic selection at codon sites, whereas FUBAR tests for evidence of passive or purifying evolution at codon sites.

## Figures and Tables

**Figure 1 toxins-15-00523-f001:**
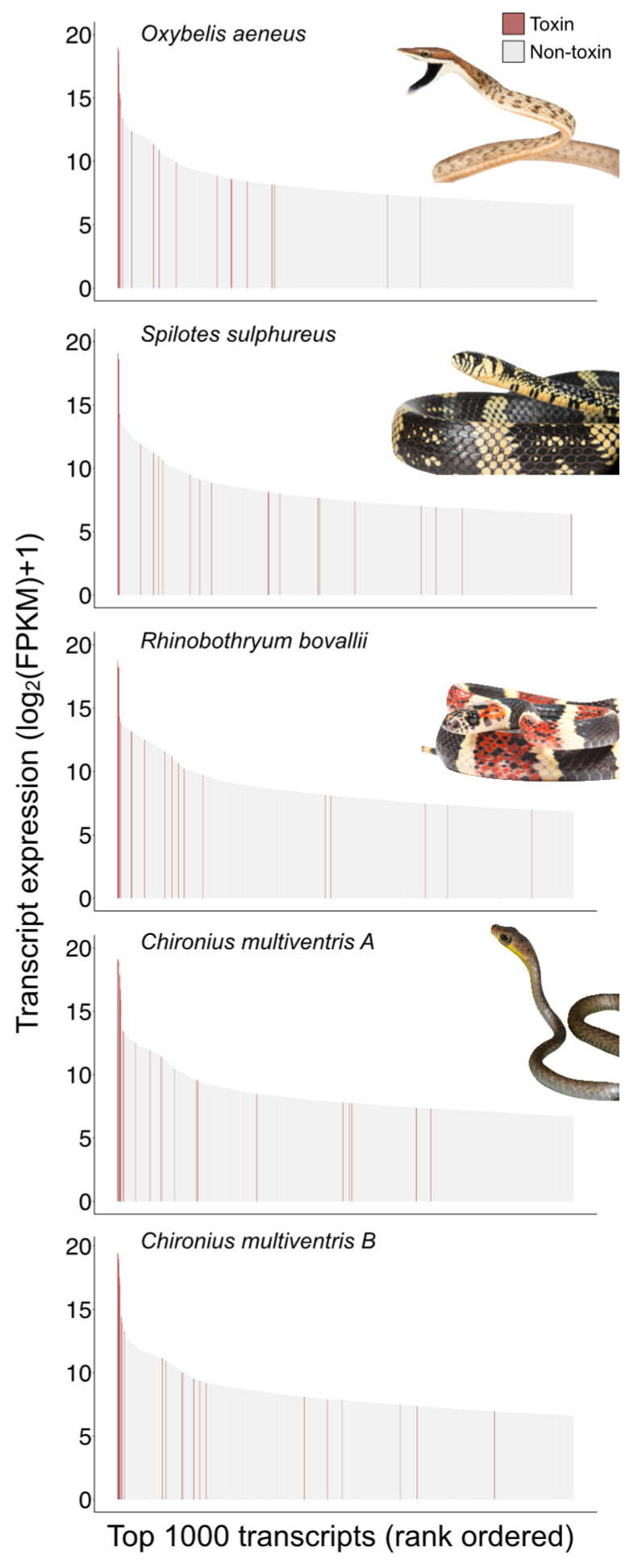
Transcript expression from the venom gland transcriptomes of four rear-fanged colubrine snake species. Red transcripts indicate toxin sequences. Transcript abundances estimates are in Fragments Per Kilobase Million (FPKM), and only the top 1000 ranked transcripts are shown. Image credits: José G. Martinez-Fonseca (*O. aeneus*, *S. sulphureus*, *R. bovallii*) and Consuelo Alarcón Rodríguez (*C. multiventris*).

**Figure 2 toxins-15-00523-f002:**
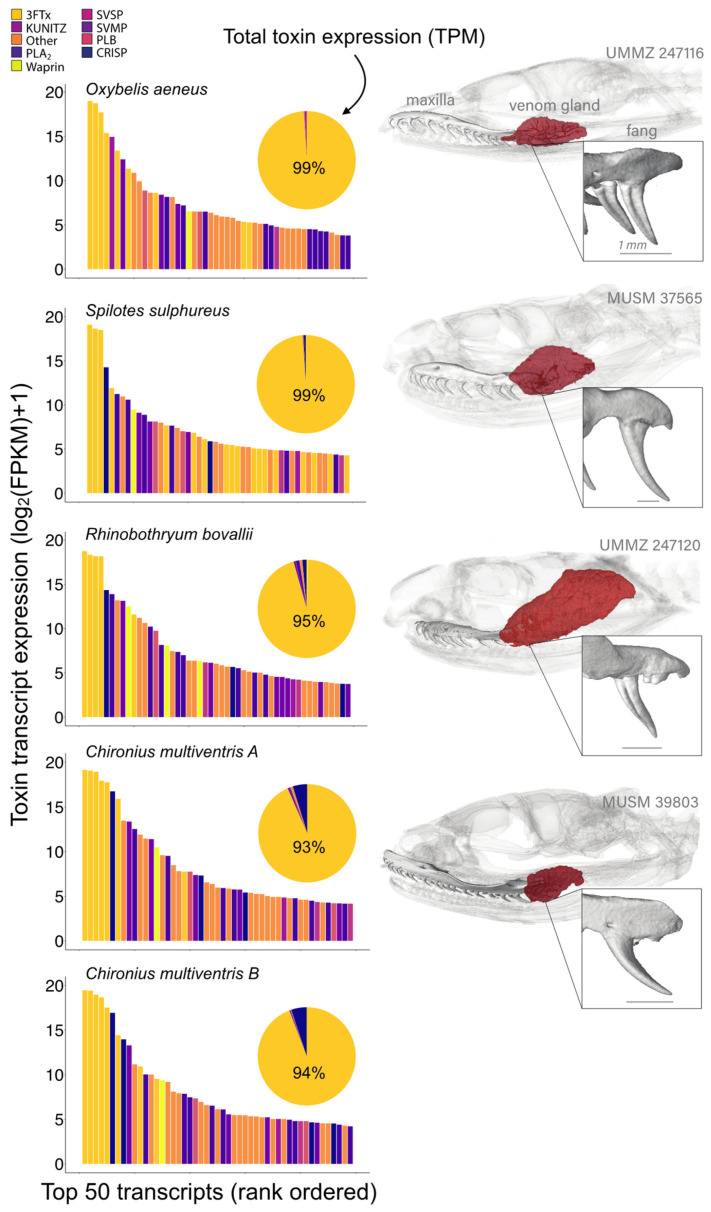
Toxin expression from the venom gland transcriptome of four species of rear-fanged colubrine snakes. Bar charts show toxin transcript abundances, measured in Fragments Per Kilobase Million (FPKM), for the top 50 toxin transcripts of each species. Pie charts show proportions of toxin family expression based on Transcripts per Million (TPM). Toxin families are color coded; toxins indicated include three-finger toxins (3FTx), Kunitz-type venom proteins (Kunitz), phospholipase A_2_ (PLA_2_), Waprin, snake venom serine proteases (SVSP), snake venom metalloproteinases (SVMPIII), L-amino acid oxidases (LAAO), phospholipases B (PLB), and Cysteine-rich secretory proteins (CRiSP). Venom system morphology is shown on the right for each species; obtained from microCT scans of specimens the University of Michigan Museum of Zoology (UMMZ) and Museo de Historia Natural de la Universidad Nacional Major de San Marcos (MUSM) in Peru. Scans are available from Morphosource (https://www.morphosource.org/Detail/ProjectDetail/Show/project_id/374).

**Figure 3 toxins-15-00523-f003:**
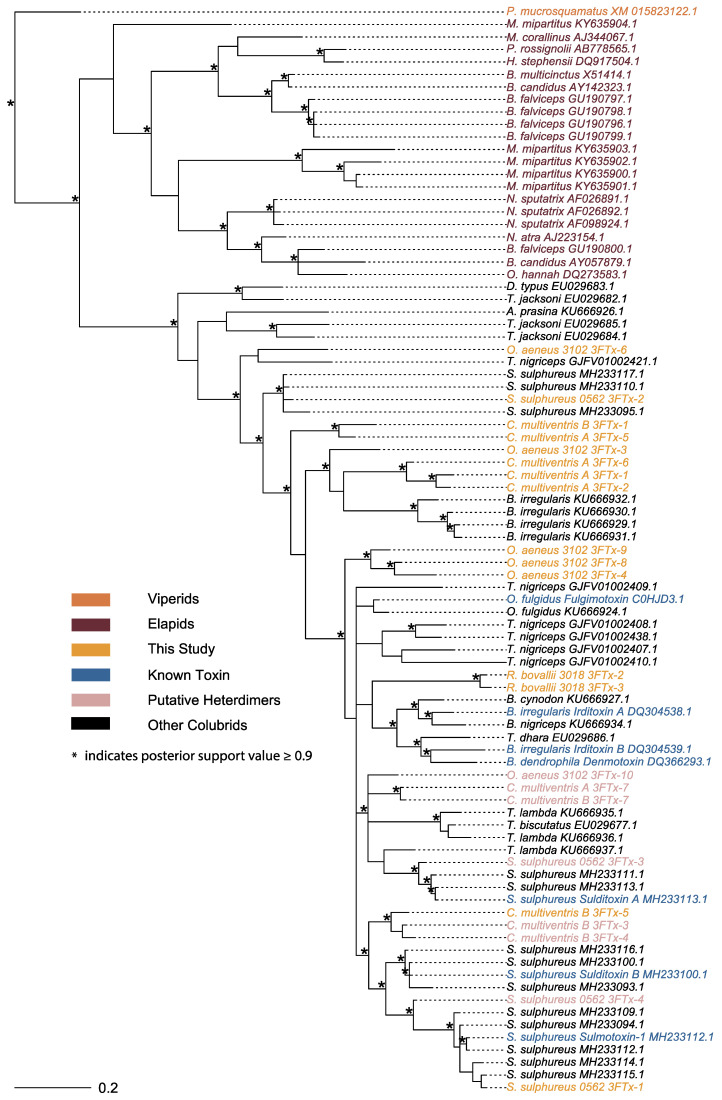
Bayesian gene tree of 3FTx sequences, including sequences newly generated in this study (gold; pink if putative heterodimer-forming sequence), previously named and functionally characterized sequences (blue), other 3FTx sequences from colubrine species (black), elapid sequences (maroon), and a sequence from a viper species used as an outgroup (orange). All 3FTx sequences not generated in this study were obtained from GenBank (Appendix A). For the species *C. multiventris*, “A” and “B” were used to differentiate between the two individuals.

**Figure 4 toxins-15-00523-f004:**
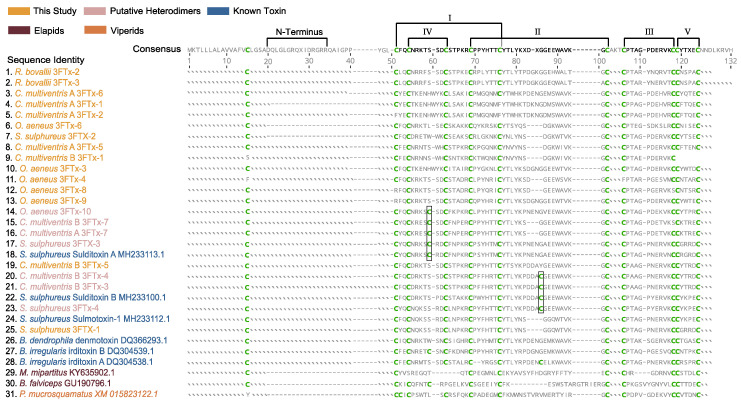
Amino acid alignment of non-conventional three-finger toxin (3FTx) transcripts generated from this study (gold; pink if putative heterodimer-forming sequence) as well as sequences previously identified and characterized from colubrine snake species (blue). Two elapid (maroon) and one viper (orange) sequence are shown as outgroups. Conserved cysteine residues are colored in green; disulfide bonds between cysteine residues result in formation of five loops in non-conventional 3FTx, which are shown with brackets. Boxes indicate sequence position of additional cysteine residues associated with heterodimer formation. Roman numerals denote the distinctive β-stranded loops of 3FTx; Loops IV and V are unique to non-conventional 3FTxs. N-terminus region indicated with bracket. All 3FTx sequences not generated in this study were obtained from GenBank (Appendix A). For the species *C. multiventris*, “A” and “B” were used to differentiate between the two individuals.

**Table 1 toxins-15-00523-t001:** Total Putative Toxin Contigs per Individual. 3FTx = Three-finger toxin; CRiSP = Cysteine rich secretory protein; PLA_2_ = Phospholipase A_2_; SVMPIII = Snake venom metalloproteinase III; and SVSP = Snake venom serine proteinase.

Taxonomy	3FTx	CRiSP	PLA_2_	SVMP	SVSP	Other
*Spilotes sulphureus*	41	16	10	29	55	60
*Rhinobothryum bovallii*	5	7	3	20	57	58
*Oxybelis aeneus*	11	6	9	9	23	38
*Chironius multiventris* A	7	21	8	27	58	53
*Chironius multiventris* B	10	25	4	34	56	55

**Table 2 toxins-15-00523-t002:** CodeML selection test comparisons. LnL = log likelihood; LRT = likelihood ratio test.

Model	np	Ln L	Estimates of Parameters			Model Compared	LRT *p*-Value
M0	62	−2330.419	ω_0_ = 1.347			M0 vs. M1	*p* < 0.001
M3	66	−2249.816	p = 0.192 ω = 0.136	0.355 0.762	0.453 3.028		
M1	63	−2280.078	p = 0.380 ω = 0.156	0.620 1		M1 vs. M2	*p* < 0.001
M2	65	−2250.100	p = 0.273 ω = 0.213	0.294 1	0.433 3.192		
M7	63	−2291.574	p = 0.051 q = 0.056			M7 vs. M8	*p* < 0.001
M8	65	−2249.800	p_0_ = 0.401 q = 0.524	p_1_ = 0.447 ω = 3.049	p = 0.652		

**Table 3 toxins-15-00523-t003:** Number of sites found under positive selection across the whole alignment, within the N-terminus region, Loop II, or Loop V. MEME = Mixed Effects Model of Evolution; FUBAR = Fast Unconstrained Bayesian Approximation.

	M2	MEME	FUBAR
Whole alignment	25	30	21
N-terminus region	3	5	3
Loop II	8	6	3
Loop V	0	4	4

**Table 4 toxins-15-00523-t004:** Specimen information. SVL = snout to vent length; MUSM = Museo de Historia Natural, Universidad Nacional Mayor de San Marcos; UMMZ = University of Michigan Museum of Zoology; EBVC = Estación Biológica Villa Carmen; RB = Refugio Bartola; EBLA = Estación Biológica Los Amigos.

Specimen	Field No.	Museum No.	SVL (mm)	Mass (g)	Capture Locality
*Spilotes sulphureus*	RAB562	MUSM 37565	1840	1250	EBVC, Peru
*Rhinobothryum bovallii*	RAB3018	UMMZ 247120	1030	71.4	RB, Nicaragua
*Oxybelis aeneus*	RAB3102	UMMZ 247113	865	73.1	Momotombo, Nicaragua
*Chironius multiventris* A	RAB3332	UMMZ 249111	810	160	EBLA, Peru
*Chironius multiventris* B	RAB3577	MUSM 39803	739	85	EBLA, Peru

## Data Availability

Tree files and protein sequence alignment are available at https://doi.org/10.7302/01w7-8d39. MicroCT scan data are available from the Morphosource repository (https://www.morphosource.org/Detail/ProjectDetail/Show/project_id/374). The raw sequence data for each venom transcriptome are available on NCBI SRA under BioProject PRJNA843733 SAMN37127315-SAMN37127319. 3FTx sequences are available on NCBI GenBank under accession numbers OR478646-OR478667. R script to plot expression data are available on GitHub (https://github.com/jcroweriddell/toxin-expression-neotropical-snakes).

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
