# Peer review of "Evolution of Three-Finger Toxin Genes in Neotropical Colubrine Snakes (Colubridae)"

_toxins, 2023, doi:10.3390/toxins15090523_

Round 1

Reviewer 1 Report

In this study authors used a transcriptomics strategy to assess the toxin diversity of rear-fanged colubrine snakes. The article is very well-written and the new findings are novel and interesting. The main objectives were clearly described and it is easy to find the associated results in the main document. Overall, this investigation is relevant and expands the current landscape of 3FTx expression profiles in among rear-fanged colubrine snakes. Thus, I only have minor comments.

1. PLA2: 2 is subscript. This has a explanation behind its use. 

2. Has the proteome of any of this species been previously studied? If so, it would be interesting to make comparisons. Are 3FTs majorities in the proteome as well?

3. In terms of PLA2, are the sequences identified from catalytically active toxins (Asp49 PLA2)? Or are catalytically inactive isoforms (Lys49 Asp 49) also identified?

4. There are different types of metalloproteases. Which type was identified in this study?

5. The authors reported differences in primary structure. Do these changes translate into changes in the 3D structure?

Author Response

We thank reviewer 1 for their feedback which improved the manuscript greatly. Our responses are below in bold.

  1. PLA2: 2 is subscript. This has a explanation behind its use. 

We revised all iterations of PLA2 to PLA2

  1. Has the proteome of any of this species been previously studied? If so, it would be interesting to make comparisons. Are 3FTs majorities in the proteome as well?

Three of the four species listed here have been previously studied while C. multiventris has not. We added two sentences to the last paragraph (lines 79-81) of the introduction to reflect this.

“Of the four species studied in this paper, 3 have had previous venom proteomic or transcriptomic studies where 3FTxs were the most abundant protein family. C. multiventris had no previous comparable studies [13, 27, 28].”

We also compare the proteomes available for these three species with our transcriptome data in the Discussion (lines 130-132). We have made this more explicit. New text is in italics.

The simple venom gene expression of 3FTx profiles of our snakes (Table S6-S10) are consistent with transcriptomic and proteomic data in previously analyzed conspecifics [13,27] and other rear-fanged colubrines [7,8,28,30].”

  1. In terms of PLA2, are the sequences identified from catalytically active toxins (Asp49 PLA2)? Or are catalytically inactive isoforms (Lys49 Asp 49) also identified?

We have identified Asp49 PLA2s in our individuals, though as the sequences were of very low expression and not the focus of this paper, we have decided not to discuss this finding in the manuscript.

  1. There are different types of metalloproteases. Which type was identified in this study?

            We have revised “SVMP” to “SVMPIII” to reflect the specific metalloproteases reviewed.

  1. The authors reported differences in primary structure. Do these changes translate into changes in the 3D structure?

Predicted 3D structure of putative 3FTx sequences from our transcriptomes match the tertiary structure of Irditoxin. We have updated the methods and results to reflect this.

Lines 242-246

“Predicted 3D protein structure and function for the colubrine 3FTxs consistently matched the two subunits of heterodimer forming irditoxin [14]. More nuanced differences in protein structure between sequences are unlikely to be predicted due to the limited published non-conventional 3FTx protein structures.”

Lines 398-404

4.6 Protein prediction

To predict secondary and tertiary structure of putative toxins transcripts, we submitted translated amino acid sequences to the I-TASSER Protein Structure and Function Predications online server (https://zhanggroup.org/I-TASSER/; [61]). This server uses a deep-learning approach by first identifying structural templates from Protein Data Bank (PDB) and then deriving functional insights by re-threading the 3D models through protein function database BioLiP [61-63].”

Reviewer 2 Report

The current study revealed the transcriptomic profiles of venom glands from four colubrine snakes in Neotropical regions, and found that 3FTx were the predominate toxins. Parallel evolution of heterodimeric interactions between 3FTxs from two snakes was further detected while integrated with the sequences from other rear-fanged colubrine snakes. Moreover, significant positive selection was detected in the relevant 3FTxs, and the number of codon sites under positive selection was also quantified. It provides new insights for understanding the venom toxin in rear-fanged snakes.

Here, I suggest the authors to conduct some necessary improvment before the acceptance of the MS.

1. line28: delete "total".

2. line77: change "RNA-sequenced" to "sequenced".

3. section2.1: The authors should indicate whether all of the 3FTxs derived in four Neotropical colubrine snakes are belonged to "non-conventional 3FTx"? 

4. line95-96: The relative abundance or expression precentage of the highest-ranking 3FTx and highest-ranking non-3FTx transcripts should be stated.

5. The authors should state the integrity (with full or partial CDS) of  all putative toxin contigs in the text.

6. Why the authors used two different parameters (FPKM and TPM) to evaluate the toxin transcript abundances?

7. line120: change "...study species..." to "snakes".

8. section2.2: The significance of parallel evolution of heterodimeric 3FTx sequences should be disscussed in-depth.

9. Table2: Why the total precentage of M1/M2/M3/M8 are all lower than 100%? Please check the precentage of M1. Please list the accurate LRT P-value and the "np" of each model.

10. The authors should explain why the MEME and FUBAR were used to predict the number of positive selected sites, rather than easyCodeML.

11. I suggest the authors to provide an additional file containing the detailed information of the positive selected codon sites. 

12. line281-282: It maybe not all of the positive selected codon sites can interact with prey targets. How to define the strength of negative selection?

13. line297: The drug and dose (or other detailed protocols) used for euthanizing the snakes should be listed.

14. line304:  How the fresh venom glands of these snakes were preserved before RNA extraction?

Author Response

We thank reviewer 2 for their feedback which improved the manuscript greatly. Our responses are below in bold.

  1. line28: delete "total".

We have incorporated the suggested change.

  1. line77: change "RNA-sequenced" to "sequenced".

We have incorporated the suggested change.

  1. section2.1: The authors should indicate whether all of the 3FTxs derived in four Neotropical colubrine snakes are belonged to "non-conventional 3FTx"? 

We have added a sentence at line 93 to indicate that none of the specimen included conventional 3FTxs. This is shown below with the new text italicized.

“The toxin transcriptomes included … No conventional 3FTxs were found among the five specimens. While many toxin transcripts…”

  1. line95-96: The relative abundance or expression precentage of the highest-ranking 3FTx and highest-ranking non-3FTx transcripts should be stated.

We edited the manuscript to include the expression (TPM) of the highest ranking 3FTx and the highest ranking non-3FTx toxin.

Lines 98-105

“The highest ranking 3FTxs had TPM values of 777,346 (O. aeneus), 735,857 (S. sulphureus, 643,746 (C. multiventris B),  571,608 (C. multiventris A), 369,994 (R. bovallii), as compared to the highest ranking non-3FTx toxin with TPM values of 5,636 (O. aeneus, KUNITZ family), 5,288 (S. sulphureus, CRISP family), 29,444 (C. multiventris A, CRISP family), 35,338 (C. multiventris B, CRISP family), and 5,659 (R. bovallii, CRISP family). See supplementary materials for total TPM and percent expression across specimens.”

  1. The authors should state the integrity (with full or partial CDS) of  all putative toxin contigs in the text.

We have added a sentence at line 203 to indicate that 30/31 sequences were full coding sequences.

The tree topology of colubrine 3FTxs shows evidence for… Of the 31 putative sequences of interest, 30 were full coding sequences (partial coding sequence C. multiventris B 3FTx-1; Figure 4).“

  1. Why the authors used two different parameters (FPKM and TPM) to evaluate the toxin transcript abundances?

We have added the following text to describe why we used two different metrics of toxin transcript abundance. New text at lines 336-369 is in italics.

We used FPKM to compare transcript abundance within individuals, as the values as the sum value is unique to each individual, and used TPM to compare transcript abundance among individuals, as the sum total is standardized at one million across samples.

  1. line120: change "...study species..." to "snakes".

We have incorporated the suggested change.

  1. section2.2: The significance of parallel evolution of heterodimeric 3FTx sequences should be disscussed in-depth.

We have expanded on the significance of parallel evolution of heterodimeric 3FTx sequences in the discussion, lines 248-260.

“Our results further support the notion that 3FTx heterodimic interactions evolved independently from the similar structure and interaction observed in Boiga snakes [11,25]. An Indomalyan-Australasian lineage within the colubrinnae radiation, Boiga cat snakes are phylogenetically distinct from the Neotropical lineages of Spilotes and Chironius, which are more closely related to each other than Boiga [3]. Other species with putative heterodimic 3FTxs cluster within the same clade as either Spilotes/Chironius (i.e. Trimorphodon, Oxybelis) or Boiga (i.e. Telescopus). This phylogenetic spread suggests that 3FTx heterodimic interactions evolved independently at least twice across the radiation of rear-fanged colubrids. However, the dearth of venom sequence data across the phylogenetic breadth of rear-fanged colubrids precludes testing this hypothesis. Further studies should aim to characterize 3FTx diversity among colubrids, especially in species with divergent prey preferences, to test the parallel evolution of heterodimic interactions for targeting taxon-specific prey.”

  1. Table2: Why the total precentage of M1/M2/M3/M8 are all lower than 100%? Please check the precentage of M1. Please list the accurate LRT P-value and the "np" of each model.

In Table 2, we have fixed the significant figure issues that resulted in p not summing up to 100%, fixed the duplicated ‘p’ for M1, and have added the ‘np’ value for each model. We have decided to keep our representation of the LRT P-value as it is effectively 0 (0.000000000), and thus have decided to represent it as p < 0.001.

  1. The authors should explain why the MEME and FUBAR were used to predict the number of positive selected sites, rather than easyCodeML.

We have included information from the result of easyCodeML’s M2 analysis to Table 3

  1. I suggest the authors to provide an additional file containing the detailed information of the positive selected codon sites. 

We have added an additional supplemental table (Table S11) containing this information and added this sentence at line 274:

“Selection test on individual sites is summarized in Table S11. ”

  1. line281-282: It maybe not all of the positive selected codon sites can interact with prey targets. How to define the strength of negative selection?

We have removed the language defining the strength of selection.

  1. line297: The drug and dose (or other detailed protocols) used for euthanizing the snakes should be listed.

We have added the following sentence at lines 331-334 to detail how animals were euthanized.

“Briefly, we euthanized the snakes with an intracoleomic 10mL dose of Xylazine followed by a 0.5-1mL intracardiac injection of Chlorobutanol after the animal has lost its righting reflex.”

  1. line304:  How the fresh venom glands of these snakes were preserved before RNA extraction?

We have changed the text to mention that the venom glands were preserved in RNALater at line 335.

“All venom glands were extracted, preserved in RNALater (Invitrogen, Carlsbad, CA, USA), and exported to the University of Michigan”

Round 2

Reviewer 2 Report

The authors carefully improved the MS, and I have no more questions.